

# Do gabapentin or pregabalin directly modulate the μ receptor?

Preeti Manandhar[1],[\*], Bridin Patricia Murnion[2],[3],[\*],
Natasha L. Grimsey[4], Mark Connor[1] and Marina Santiago[1]

[1] Department of Biomedical Sciences, Macquarie University, Sydney, NSW, Australia
[2] Drug and Alcohol Services, Central Coast Local Health District, Hamlyn Terrace, NSW, Australia
[3] School of Medicine and Public Health, University of Newcastle, Newcastle, NSW, Australia
[4] Department of Pharmacology and Clinical Pharmacology, University of Auckland, Auckland, New Zealand
[\*] These authors contributed equally to this work.

Corresponding author
Marina Santiago,
marina.junqueirasantiago@mq.edu.au

## ABSTRACT

**Background:** Pregabalin and gabapentin improve neuropathic pain symptoms but there are emerging concerns regarding their misuse. This is more pronounced among patients with substance use disorder, particularly involving opioids. Co-ingestion of gabapentinoids with opioids is increasingly identified in opioid related deaths, however, the molecular mechanism behind this is still unclear. We have sought to determine whether pregabalin or gabapentin directly modulates acute μ receptor signaling, or μ receptor activation by morphine.

**Methods:** The effects of pregabalin and gabapentin were assessed in HEK 293 cells stably transfected with the human μ receptor. Their effect on morphine induced hyperpolarization, cAMP production and ERK phosphorylation were studied using fluorescent-based membrane potential assay, bioluminescence based CAMYEL assay and ELISA assay, respectively. Pregabalin/gabapentin effects on morphine-induced hyperpolarization were also investigated in AtT20 cells.

**Results:** Pregabalin or gabapentin (1 μM, 100 μM each) did not activate the μ receptor or affect K channel activation or ERK phosphorylation produced by morphine. Neither drug affected the desensitization of K channel activation produced by prolonged (30 min) application of morphine. Gabapentin (1 μM, 100 μM) and pregabalin (1 μM) did not affect inhibition of forskolin-stimulated cAMP production by morphine. However, pregabalin (100 μM) potentiated forskolin mediated cAMP production, although morphine still inhibited cAMP levels with a similar potency to control.

**Discussion:** Pregabalin or gabapentin did not activate or modulate μ receptor signaling in three different assays. Our data do not support the hypothesis that gabapentin or pregabalin augment opioid effects through direct or allosteric modulation of the μ receptor. Pregabalin at a high concentration increases cAMP production independent of morphine. The mechanism of enhanced opioid-related harms from co-ingestion of pregabalin or gabapentin with opioids needs further investigation.

## INTRODUCTION

Gabapentin and pregabalin improve analgesia in acute perioperative pain and chronic neuropathic pain (*Finnerup et al., 2015*; *Schug et al., 2015*). They are commonly used as an opioid-sparing agent (*Freedman & O'Hara, 2008*; *Tiippana et al., 2007*), particularly in opioid-tolerant patients (*Simpson & Jackson, 2017*). Despite clinical benefit, there are increasing reports of abuse of these medications (*Cairns et al., 2019*). Gabapentin and pregabalin have been reported to enhance the psychoactive effect of opioids by people who use the drugs in an unregulated fashion (*Baird, Fox & Colvin, 2014*; *Lyndon et al., 2017*). Coronial post-mortem data identifies the appearance of pregabalin or gabapentin in an increasing proportion of opioid-related fatalities (*Cairns et al., 2019*; *Lyndon et al., 2017*).

While both gabapentin and pregabalin are GABA analogs, no direct action at GABA receptors has been identified (*Lanneau et al., 2001*; *Jensen et al., 2002*; *Stahl et al., 2013*). The analgesic effect is thought to be a result of these drug's interaction with the alpha-2-delta subunit of the presynaptic voltage-gated calcium channel (*Chincholkar, 2018*). However, multiple potential mechanisms of action have been reported, including interaction with L-amino transport system, action at N-methy-D-aspartate (NMDA) receptors, blockade of voltage-gated sodium channels and increased conductance of potassium channels (*Chincholkar, 2018*; *Manville & Abbott, 2018*; *Taylor & Harris, 2020*).

The μ receptor (previously known as μ opioid receptor) and its endogenous ligands are crucial regulators of cells and circuits that detect, integrate, and respond to painful sensations. Clinically used opioids exert their analgesic action through activation of the μ receptor (*Al-Hasani & Bruchas, 2011*), however, a few such as tapentadol and tramadol may also modulate other pain-related targets (*Faria et al., 2018*). More recently, the analgesic potential of allosteric modulators of the μ receptor have been investigated (*Burford, Traynor & Alt, 2015*). During an acute pain episode, a positive allosteric modulator could enhance endorphin effect at the μ receptor on pain modulating sites; however, it is expected to have minimal off-target effect as in other areas endogenous ligands may not be present to activate the μ receptor.

Animal studies suggest gabapentinoid activity in modulating neuropathic pain is opioid independent (*Kremer et al., 2016*). However, different mechanisms may underpin opioid-sparing, reward, and respiratory depression/sedation, with the latter properties leading to misuse and toxicity. Pregabalin has been shown to reverse the tolerance to morphine-induced respiratory depression in mice (*Lyndon et al., 2017*) and potentiation of morphine signaling through allosteric modulation of the μ receptor could account for this effect.

To date, there has been no investigation of the actions of gabapentin or pregabalin at the μ receptor. We, therefore, hypothesized that gabapentin and pregabalin enhance opioid effect through direct modulation of the μ receptor. The current study investigated

recombinant human μ receptor modulation of three distinct effector systems: activation of G protein-coupled inwardly rectifying potassium channels (GIRK), inhibition of adenylyl cyclase, and stimulation of extracellular signal-regulated kinase (ERK) phosphorylation.

# MATERIALS AND METHODS

## Cell culture

The study was performed in Human Embryonic Kidney Flp-In™-293 (HEK) cells from Life Technologies (R75007; Thermo Fisher Scientific, Waltham, MA, USA, derived from ATCC CRL-1573) and mouse pituitary (AtT20) cells from ATCC (CRL-1795). HEK cells were stably transfected with human GIRK 4 construct subcloned into pcDNA3.1$^+$ (*Gillis et al., 2020*) and 3-haemagglutinin tagged human μ receptor construct subcloned into pcDNA5/FRT/TO (HEK-μ). AtT20 Flp-In™ cells were stably transfected with the same human μ receptor-pcDNA5/FRT/TO (AtT20-μ) as described previously (*Knapman et al., 2013*) using Fugene HD. The integration of GIRK-4-pcDNA3.1$^+$ plasmids led to geneticin resistant cells and a single cell was expanded to produce a clonal cell line. Human μ receptor-pcDNA5/FRT/TO was cotransfected with pOG44 into the Flp-In™ site according to manufacturer's instructions (Thermo Fisher Scientific, Waltham, MA, USA). Selection was performed using 150 μg/mL hygromycin gold and resistant cells are considered isogenic due to integration of the plasmid into the same genomic locus in every clone (Flp-In™ system).

Cells were cultured and maintained in growth media composed of Dulbecco's modified Eagle's medium (DMEM) supplemented with 10% fetal bovine saline (FBS), 100 U/mL penicillin, 100 μg/mL streptomycin, and 80 μg/mL hygromycin gold. HEK cells additionally had 400 μg/mL geneticin added to the culture media. All cells were stored in a humidified incubator at 37 °C in an atmosphere with 5% $CO_2$.

## Hyperpolarization assay

The effect of pregabalin or gabapentin on morphine induced hyperpolarization was evaluated using a fluorescence-based membrane potential assay (*Knapman et al., 2013*). One day before the assay, cells were detached from the flask with trypsin/EDTA and resuspended in Leibovitz (L-15) supplemented with 1% FBS, 100 U/mL penicillin, 100 μg/mL streptomycin and 15 mM glucose. A total of 90 μL cell suspension was seeded onto poly-D-lysine coated 96 well black-walled clear bottomed plates to achieve a monolayer and incubated overnight at 37 °C and ambient $CO_2$.

FLIPR membrane potential blue dye was reconstituted to 50% of the manufacturer's recommended concentration in modified Hank's buffered salt solution (HBSS). This was composed of (in mM) NaCl 145, Na$_2$HPO4 0.338, NaHCO$_3$ 4.17, HEPES 22, KH$_2$PO$_4$ 0.441, MgSO$_4$ 0.407, MgCl$_2$ 0.493, CaCl$_2$ 1.26, glucose 5.56 (pH 7.4, osmolarity 315 ± 15). A total of 90 μL of dye was added to each well and incubated for 60 min at 37 °C in ambient $CO_2$ prior to fluorescence reading. All drugs were diluted in HBSS and loaded on the compound plate. Drug concentrations were ten times that of the final concentration to accommodate for dilution after addition to the well. Fluorescence was measured using

FlexStation 3 plate reader (Molecular Devices) with an excitation wavelength of 530 nm and emission of 565 nm at 2 s intervals.

Baseline fluorescence was recorded for 2 min after which 20 µL of vehicle or drug was added. Data were calculated as a percentage change from baseline fluorescence after correction for vehicle addition.

## Assay for cAMP measurement

Bioluminescence based cAMP assay was performed in HEK-µ cells to investigate the effect of pregabalin or gabapentin on morphine mediated inhibition of cAMP production (*Sachdev et al., 2019*; *Manandhar, Sachdev & Santiago, 2020*). The cells for the assay were detached from the flask with trypsin/EDTA and seeded in a 10 cm plate at a density of 6 million cells to reach 60–70% confluency in 24 h. Next day, pcDNA3L-His-CAMYEL plasmid was transiently transfected into the cells using linear polyethyleneimine (PEI) at a ratio of 1:6 (DNA: PEI). The transfected cell plate was then incubated overnight at 37 °C in a humidified atmosphere with 5% $CO_2$. A total of 24 h after transfection, the cells were detached from the plate and resuspended in L-15 (no phenol red), supplemented with 1% FBS, 100 U/mL penicillin, 100 µg/mL streptomycin and 15 mM glucose. Cells were then plated in 96 well white-walled clear bottom plates pre-coated with poly-D-lysine and incubated overnight at 37 °C in ambient $CO_2$.

On the day of the assay, all drugs were prepared at ten times the final concentration in HBSS containing 30 µM forskolin to reach the desired final concentration upon adding to the cell suspension. Cellular cAMP levels were measured using kinetics settings in the FlexStation 3 and luminescence detection was set at an emission spectrum of 461 nm (Rluc) and 542 nm (YFP). A total of 10 µL coelenterazine-h substrate was added, and the baseline luminescence was measured for 5 min before adding the drugs. Inverse BRET ratio was calculated at the emission of 461/542 so that the increase in ratio corresponds to increase cAMP production. The area under the curve was calculated for each concentration of the drug after correcting for vehicle and data were expressed as a percentage change in forskolin response after drug addition.

## Whole cell ELISA for measuring ERK phosphorylation

Opioid-induced ERK phosphorylation was measured using whole cell ELISA in HEK-µ cells. The cells were resuspended in L-15 and plated in a 96 well clear microplate pre-coated with poly-D-lysine and incubated overnight at 37 °C. The next day, cells were serum starved for an hour in serum free L-15 before adding the drugs. Cells were then treated for 5 min with different concentrations of drugs prepared in serum free L-15. Some wells were separated for treatment with the positive control (100 nM PMA, Phorbol 12-Myristate 13-Acetate) and negative control (10 µM U0126). After 5 min of incubation, drug solution was removed, and the plate was immediately placed on ice. Cells were fixed using 4% paraformaldehyde for 15 min at room temperature. The cells were washed 3 times with PBS then permeabilized with 0.1% Triton-X in PBS. After 30 min, the permeabilizing agent was removed and the plate was incubated at room temperature for 2 h with blocking solution (5% bovine serum albumin (BSA) in PBS with 0.01%

Tween-20). After 2 h of blocking, the solution was removed, and cells were incubated overnight at 4 °C with rabbit α-phospho-p44/42 MAPK (Thr202/Tyr204) antibody diluted at a ratio of 1:500 with 1% BSA in PBS supplemented with 0.1% Tween-20 (PBS-T). Next day the cells were washed 3 times with washing buffer (PBS-T) and treated with secondary antibody (1:5,000 α-rabbit IgG HRP-linked antibody in PBS-T containing 1% BSA) for 2 h. Cells were washed three times and incubated in the dark with ELISA peroxidase substrate (3,3′,5,5′-tetramethylbenzidine) for 45 min. Finally, the reaction was stopped using 1N HCl. Absorbance was measured at 450 nm using the BMG PheraStar FS plate reader. Data were expressed as mean ± SEM for five individual experiments performed in triplicates and the concentration-response curve was plotted after normalizing to the positive control.

### Data analysis

All data were expressed as mean ± SEM of at least five independent experiments performed in duplicates unless otherwise stated. Concentration-response curves were plotted using the four-parameter non-linear regression equation in GraphPad PRISM 8 (GraphPad Software, San Diego, CA, USA). Data were statistically tested using one way ANOVA corrected using Bonferroni method within each set of comparisons. Two tailed unpaired student t-test was used to compare the mean of $EC_{50}$ and $E_{max}$ for CAMYEL assay in measuring the effect of 100 μM pregabalin and isoproterenol in forskolin response. $P$ values < 0.05 were considered significant for all the analysis.

## Materials

Morphine was a kind gift from the University of Sydney, Australia. Pregabalin and gabapentin were purchased from Tocris, Australia. Forskolin was provided by Ascent Scientific Ltd. Culture reagents and buffer salts were supplied by Sigma Aldrich (Castle Hill, Australia) or Thermo Fisher Scientific (Waltham, MA, USA). The CAMYEL plasmid encoding the cAMP sensor YFP-EPac-RLuc was originally from ATCC (MBA-277). Phospho-ERK1/2 antibody and anti-rabbit IgG HRP linked antibody were from Cell Signaling Technologies (Danvers, MA, USA). Antibiotics were from Invivogen (San Diego, CA, USA) and coelenterazine h was from Promega (Alexandria, Australia). The membrane potential dye was purchased from Molecular Devices, CA.

## RESULTS

The effects of pregabalin and gabapentin on morphine-induced K channel activation was investigated in HEK cells co-expressing GIRK4 and human μ receptor. Morphine caused maximum hyperpolarization at 13 ± 4% with pEC50 of 6.6 ± 0.5. Incubation of pregabalin or gabapentin at 1 μM or 100 μM for 5 min did not change the morphine-induced GIRK activation in HEK-μ cells (Figs. 1A, 1C and 1E). $E_{max}$, pEC50 and $p$ values are shown in Table 1.

To ensure the effect was not cell-specific, the hyperpolarization response to morphine with and without pregabalin or gabapentin was assessed in AtT20-μ cells. While the maximum hyperpolarization caused by morphine was higher in AtT20-μ cells as

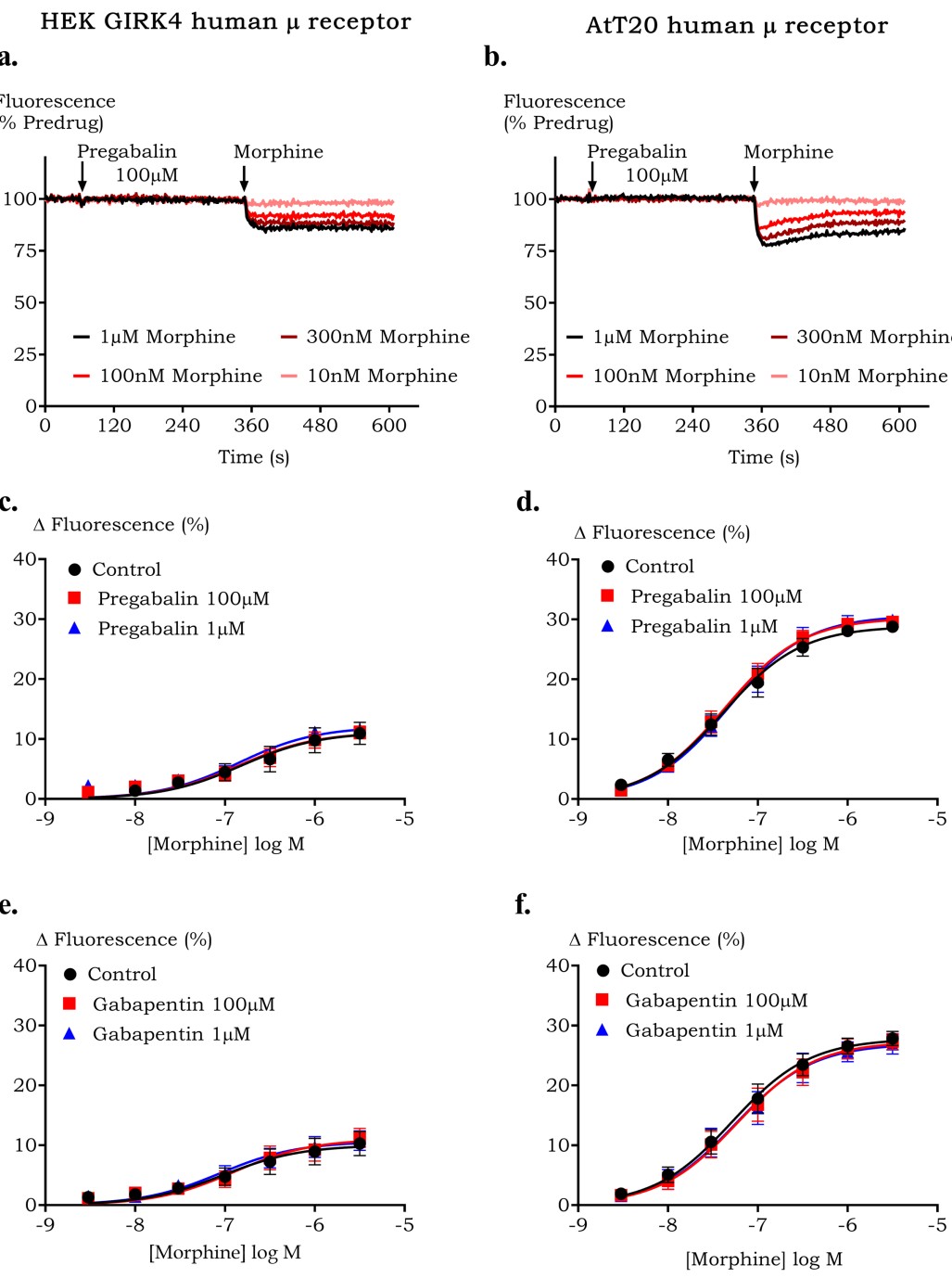

**Figure 1 The effects of pregabalin or gabapentin in morphine induced GIRK channel activation.** Traces showing changes in fluorescent signal for various concentrations of morphine following application of 100 μM pregabalin in (A) HEK 293-μ cells and (B) AtT20-μ cells. Drugs are added on the time as pointed by arrow and values are normalized to baseline reading. Concentration response-curves of morphine induced hyperpolarization of HEK 293-μ cells with vehicle, (C) pregabalin (100 μM or 1 μM) or (D) gabapentin (100 μM or 1 μM) and morphine-induced hyperpolarization of AtT20-μ cells with vehicle, (E) pregabalin (100 μM or 1 μM) or (F) gabapentin (100 μM or 1 μM) (SEM, $n \geq 5$).

**Table 1 Activity of morphine with or without pregabalin or gabapentin in membrane potential assay in HEK and AtT20 cells expressing μ receptor. Summary of curve-derived parameters.**

| | HEK cells | | | | AtT20 cells | | | |
|---|---|---|---|---|---|---|---|---|
| | $E_{max}$ | p value | pEC50 | p value | $E_{max}$ | p value | pEC50 | p value |
| Control | 12.68 ± 4% | | 6.6 ± 0.5 | | 29.76 ± 1% | | 7.3 ± 0.1 | |
| 100 μM pregabalin | 15.47 ± 6 % | >0.99 | 6.3 ± 0.7 | >0.99 | 30.12 ± 1% | >0.99 | 7.4 ± 0.1 | >0.99 |
| 1 μM pregabalin | 17.76 ± 6% | 0.78 | 6.2 ± 0.7 | >0.99 | 30.9 ± 1% | 0.17 | 7.3 ± 0.1 | >0.99 |
| 100 μM gabapentin | 13.93 ± 5% | 0.13 | 6.5 ± 0.7 | 0.36 | 28.19 ± 2% | >0.99 | 7.2 ± 0.1 | >0.99 |
| 1 μM gabapentin | 12.36 ± 3% | >0.99 | 6.8 ± 0.4 | >0.99 | 27.72 ± 2% | >0.99 | 7.3 ± 0.1 | >0.99 |

compared to HEK-μ cells, incubation with pregabalin or gabapentin did not affect the response to morphine (Figs, 1B, 1D and 1F). The corresponding $E_{max}$, pEC50 and p values are shown in Table 1.

Modulation of agonist dependent acute desensitization was investigated by incubating cells for 5 min with 100 μM pregabalin or gabapentin, followed by the addition of 10 μM morphine and fluorescence monitoring for 30 min (Fig. 2). Acute desensitization was measured as change in area under the curve after addition of pregabalin/gabapentin or vehicle. Pregabalin or gabapentin did not change acute desensitization of μ receptor in both AtT20-μ ($p =$ >0.99 for pregabalin and $p = 0.88$ for gabapentin) or HEK-μ cells ($p =$ >0.99 for pregabalin and gabapentin). In HEK-μ cells, membrane hyperpolarization in response to morphine, although lower in amplitude, persisted with minimal abatement over the 30 min it was measured (Fig. 2A). In comparison, morphine response was greater in magnitude in AtT20-μ cells but showed attenuation over 30 min (Fig. 2B).

To investigate the effect of prolonged pre-treatment with gabapentin/pregabalin on the opioid response in HEK-μ cells, the morphine concentration-response curve was performed after 60 min incubation with 100 μM pregabalin or gabapentin. There was no difference in the morphine response in all three groups, with maximum decrease in fluorescence at 16 ± 2% in control and 15 ± 2% ($p = 0.84$) and 16 ± 3% ($p =$ >0.99) in the presence of 100 μM pregabalin and gabapentin respectively, with a constant potency value of 6.7 ± 0.3 in all three conditions (Fig. 3).

The effect of pregabalin/gabapentin on morphine response was further evaluated in a bioluminescence-based forskolin-stimulated cAMP assay in HEK cells. One μM pregabalin, 1 and 100 μM gabapentin co-administered with forskolin did not modulate the changes in cAMP levels mediated by forskolin alone or forskolin with morphine. Morphine inhibited the forskolin-stimulated increase in cAMP by 41± 4% with pEC50 of 6.8 ± 0.2. In the presence of 1 μM pregabalin, 1 μM or 100 μM gabapentin, maximum inhibition of cAMP by morphine was 37 ± 3% ($p =$ >0.99), 47 ± 5% ($p =$ >0.99) and 35 ± 4% ($p = 0.95$) respectively with corresponding pEC50 values as 6.9 ± 0.1, 6.8 ± 0.2 and 7.0 ± 0.2 (Fig. 4). However, 100 μM pregabalin co-applied with forskolin augmented the cAMP levels compared to forskolin alone (Fig. 4D). The assay was repeated with
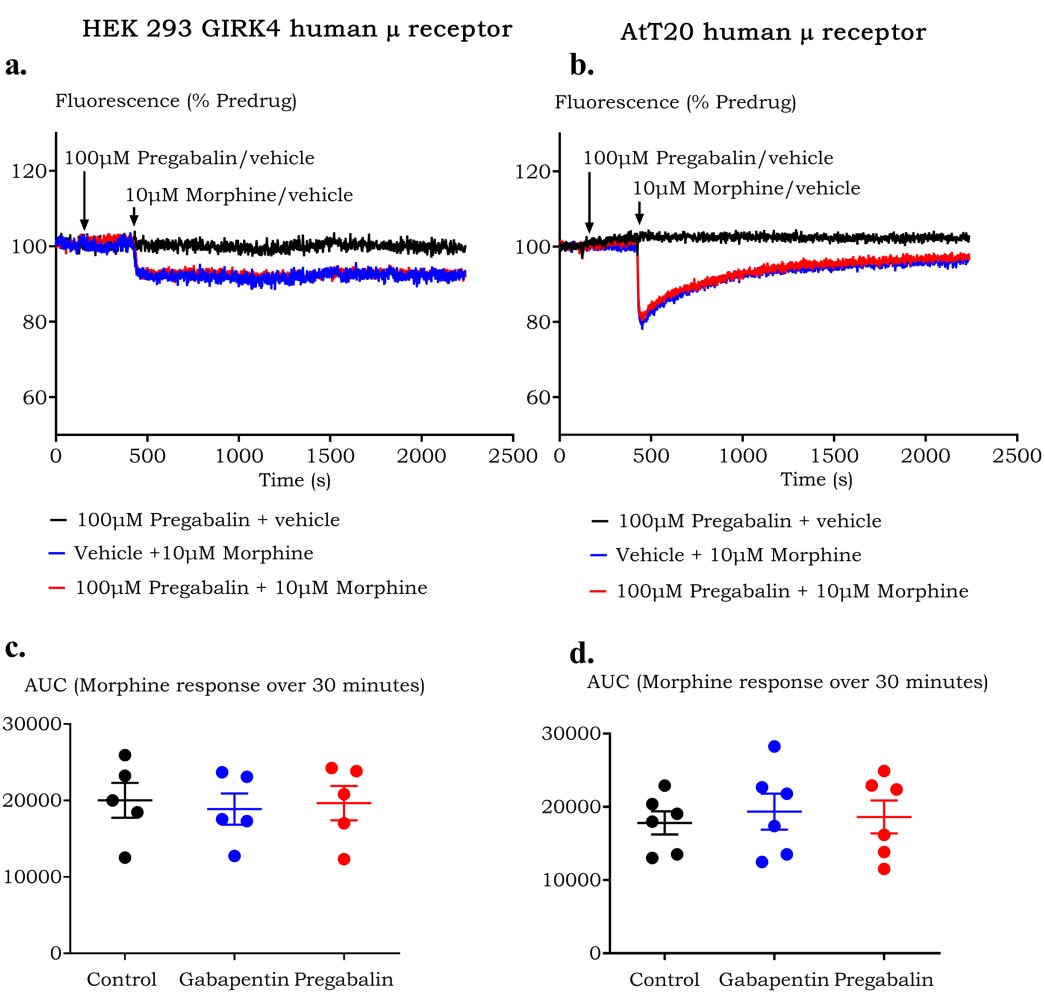

**Figure 2 Morphine induced signal desensitization is not modulated by the presence of pregabalin or gabapentin in HEK 293 and AtT20 cells expressing human μ receptors.** Raw traces showing desensitization of fluorescence signals in (A) HEK 293-μ cells and (B) AtT20-μ cells on prolonged treatment with 10 μM morphine after pre-incubation with vehicle (HBSS) or 100 μM pregabalin. Scattered dot plots showing area under the curve after prolonged stimulation by morphine after exposure to pregabalin or gabapentin or vehicle in (C) HEK 293-μ cells and (D) AtT20-μ cells (SEM, $n \geq 5$). There was no difference in the maximum effect or potency of morphine-induced desensitization in HEK or AtT20 μ cells.

concentrations of pregabalin between 10 μM and 10 nM. This identified pregabalin concentrations $\geq 100$ μM cause an increase in cAMP when co-applied with forskolin.

To test the hypothesis that this increase may be due to activation of $G\alpha_s$, the concentration-response curve of forskolin and isoproterenol (a $G\alpha_s$ agonist) in the presence of 100 μM pregabalin in HEK GIRK4 wild type cells were plotted. Pregabalin did not affect the maximum response of forskolin but caused a significant increase in potency of forskolin (Figs. 4E, $p = 0.015$), however, it did not affect the response of isoproterenol (Fig. 4F).

Finally, the effect of gabapentin or pregabalin on the ligand interaction of μ receptor on ERK phosphorylation pathway was investigated using ELISA. Acute application of 100 μM

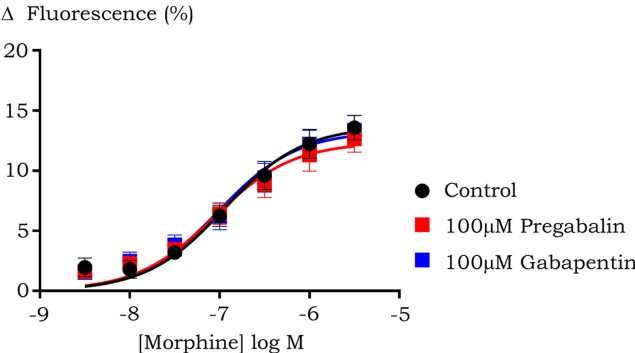

**Figure 3 Concentration-response curve of morphine induced hyperpolarization in HEK 293-µ cells after prolonged (60 min) pre-treatment with pregabalin or gabapentin.** Prolonged pre-incubation did not affect hyperpolarization induced by morphine. Data are expressed as percentage of change in fluorescence from baseline after correction for vehicle (SEM, $n \geq 5$).

pregabalin ($p > 0.99$) and 100 µM gabapentin ($p > 0.99$) did not affect the phosphorylation of ERK on HEK µ cells. We found that morphine $E_{max}$ for ERK1/2 phosphorylation was 28 ± 2% of the PMA response with $p$EC50 value of 6.9 ± 0.2. There was no significant change in the morphine-induced phosphorylation in the presence of 1 µM or 100 µM of either pregabalin or gabapentin (Fig. 5). Morphine caused maximum phosphorylation of 21 ± 4% ($p = >0.99$) and 22 ± 4% ($p = >0.99$) with 1 µM or 100 µM pregabalin with $p$EC50 of 6.6 ± 0.4 ($p = 0.31$) and 6.5 ± 0.4 ($p = 0.23$), respectively. Co-administration of 1 µM or 100 µM gabapentin with morphine resulted in maximum phosphorylation of 30 ± 4% ($p = 0.73$) and 27 ± 4% ($p = 0.92$) respectively with a $p$EC50 value of 6.7 ± 0.3 ($p = >0.99$)

# DISCUSSION

In this study, we have shown that neither pregabalin nor gabapentin affects morphine signaling at the µ receptor when modulation of GIRK activation, ERK phosphorylation or cAMP production was measured using in vitro assays. No direct effect of pregabalin or gabapentin on GIRK activation or ERK phosphorylation was observed. Neither gabapentin nor lower concentrations of pregabalin impacted forskolin-stimulated cAMP, nor did it alter the inhibition of forskolin-stimulated cAMP produced by morphine. Interestingly, 100 µM pregabalin enhanced forskolin-stimulated cAMP, which has not previously been demonstrated. Finally, the presence of pregabalin and gabapentin did not affect acute desensitization of GIRK activation in response to prolonged exposure to morphine. These data suggest that direct interaction of pregabalin and gabapentin with the µ receptor is unlikely to explain its use as an adjuvant to opioids in situations of unregulated use, and any positive effects of pregabalin and gabapentin on opioid user experience may be mediated through interactions with the cells and circuits that are modulated by µ receptors.

Pregabalin concentration at its site of action is not known, however, a single 300 mg dose of pregabalin in peri-operative patients can result in cerebrospinal fluid (CSF)

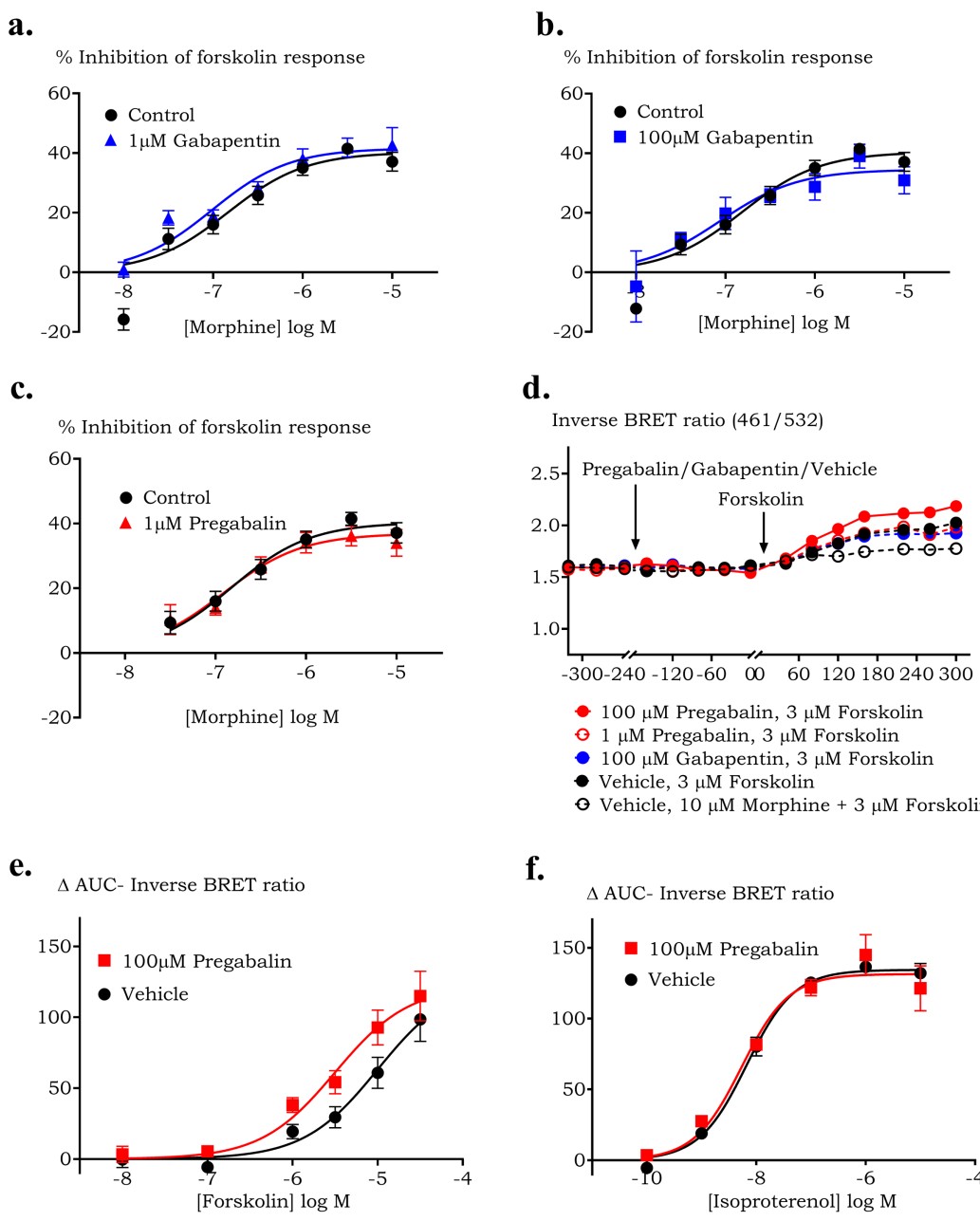

**Figure 4** **The effect of pregabalin or gabapentin on morphine induced cAMP production in HEK 293 GIRK4 cells expressing human μ receptor.** Concentration-response curves of morphine's response to forskolin-stimulated cAMP production in the presence of (A) 1 μM gabapentin, (B) 100 μM gabapentin or (C) 1 μM pregabalin. Representative traces showing the effect of 100 μM pregabalin on forskolin-stimulated increase in cAMP levels in HEK-μ cells (D). An increase in BRET ratio (461/542 nm) corresponds to increase in cAMP production. (E) A total of 100 μM pregabalin increased the potency to inhibit forskolin-stimulated increase in cAMP; however, it did not change the response of (F) isoproterenol stimulated cAMP accumulation.

concentrations of approximately 0.4 μg/mL (2.5 μM) (*Buvanendran et al., 2010*). A total of 100 μM pregabalin is also higher than measured in plasma after single dosing (12.5–50 μM) or steady-state after multiple dosing (80–90 μM) (*Bockbrader et al., 2010b*;

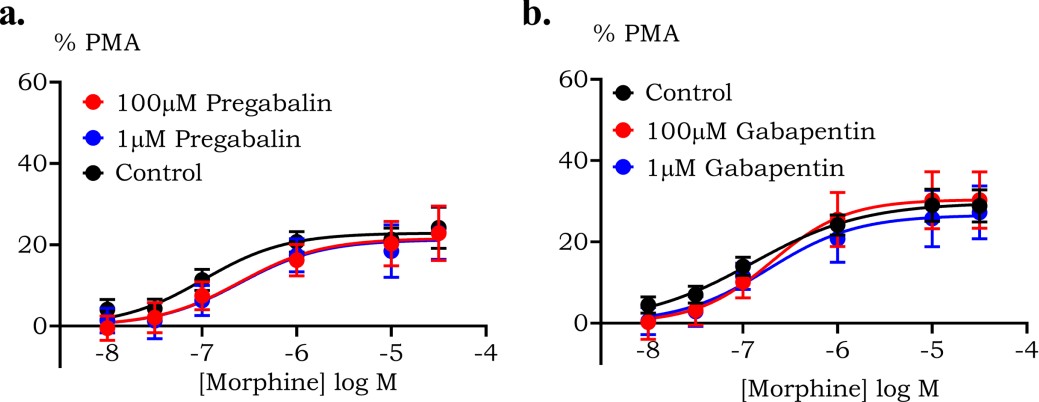

**Figure 5 Pregabalin or gabapentin does not modulate morphine induced ERK phosphorylation in HEK 293 cells expressing human μ receptor.** Levels of ERK phosphorylation was assessed using ELISA assay as described in methods. Data are calculated as percentage change from 100 nM PMA (positive control). Morphine stimulated ERK1/2 phosphorylation to a similar degree in the presence or absence of (A) 1 μM or 100 μM pregabalin or (B) 1 μM or 100 μM gabapentin (SEM, $n \geq 6$ performed in triplicate).

*May et al., 2007*). However, the above concentrations were measured in patients taking pregabalin at the recommended clinical dosage, which is not necessarily relevant when these drugs are misused. It is evident that those with substance use disorder ingest large doses of pregabalin (*WHO, 2018*). In a two-year study of post-mortem blood pregabalin concentration, it was observed that one-third of the seventy analyzed samples were above 17 mg/L (106 μM) (*Eastwood & Davison, 2016*). In a study of drug-affected drivers, pregabalin plasma concentration ranged from 0.68 to 111.6 mg/L (3.8–700 μM) (*Kriikku et al., 2014*). Concentration of gabapentin detected in CSF after a single dose of 600–1,200 mg ranges between 0.15 μg/mL and 0.44 μg/mL, equating to 0.9-2.6 μM (*Ben-Menachem, Persson & Hedner, 1992*). The 1 μM concentrations of both pregabalin and gabapentin used in our assays are therefore broadly similar to those expected in CSF in routine clinical practice. Likewise, the higher concentrations are consistent with reports associated with drug intoxication, at least for pregabalin. Activation of a novel signaling pathway at higher concentrations could contribute to toxicity seen at these higher doses.

The finding that higher dose pregabalin augmented forskolin-stimulated cAMP activity with no effect from gabapentin also identifies a previously unrecognized difference in the drugs. Pregabalin and gabapentin are often considered interchangeable clinically. There are some differences in pharmacokinetic parameters, such as dose-dependent (non-linear) absorption and lower bioavailability of gabapentin (*Bockbrader et al., 2010a*). Looking at human efficacy data, pregabalin shows a dose-dependent improvement in neuropathic pain, while gabapentin shows no dose-response relationship (*Finnerup et al., 2015*). The number needed to harm in clinical trials is lower for pregabalin than gabapentin (*Finnerup et al., 2015*), and heroin users indicate a preference for pregabalin (*Lyndon et al., 2017*). We have demonstrated that higher concentration of pregabalin increased cAMP levels which was not seen with gabapentin. The increment in cAMP possibly being independent of $G\alpha_s$ and μ receptor being activated by Gi/Go, highlight the

need of more studies to better understand this unknown mechanism and if it has any link to the synergistic toxicity seen with opioid and pregabalin.

As the lethality of pregabalin/gabapentin combined with opioids is increasingly recognized, so has the need to understand the molecular biology of this toxicity. *Lyndon et al. (2017)* identified that pregabalin alone causes respiratory depression that is not blocked by naloxone, and therefore not an opioid-mediated effect. Pregabalin binding to an allosteric site on the μ receptor is still a possibility but our data concur with *Lyndon et al. (2017)* in that no direct effect on the μ receptor signaling was seen with pregabalin and gabapentin. In the same study, Lyndon noted opioid-tolerant mice lose their tolerance to opioid-induced respiratory depression when pregabalin is administered along with morphine. Acute desensitization is arguably an early step for the development of tolerance (*Borgland, 2001*). Our study demonstrated no changes in acute desensitization to morphine with or without pregabalin or gabapentin, suggesting that any changes in tolerance associated with pregabalin is not linked to a change in morphine-mediated acute desensitization of the μ receptor. However, many kinases have been shown to be involved in μ receptor acute desensitization which leaves open the possibility of pregabalin and gabapentin affecting desensitization by mechanism(s) activated by other opioids or by pathways not present in the cell lines tested in this study. Considering the results presented in this study, the effect of pregabalin in tolerance is more likely to be an indirect effect through common pathways.

Both positive and negative allosteric modulators of the μ receptor have been identified (*Burford, Traynor & Alt, 2015*). Recent data suggest that allosteric ligands may have effects at multiple opioid receptor types, and an allosteric binding site that is conserved across opioid receptors may account for this finding (*Livingston et al., 2018*). This study did not demonstrate any augmentation or diminution of morphine signaling with either pregabalin or gabapentin. It is therefore unlikely that these compounds act as allosteric modulators of the μ receptor. Our experiments were conducted with only one orthosteric ligand, so agonist (probe) dependance has not been excluded. However, the fact that morphine with pregabalin or gabapentin in vivo have synergistic effect argue against this (*Turan et al., 2004*; *Keskinbora, Pekel & Aydinli, 2007*).

Performing the experiments in different cell lines strengthens the robustness of the findings, hence we used two different cell lines. Although HEK cells showed a lower maximum response and minimal acute signal desensitization in response to prolonged exposure to morphine as compared to AtT20 cell, pregabalin or gabapentin did not alter morphine response in either HEK or AtT20 cells, which supports our findings.

## CONCLUSIONS

In conclusion, this is the first study to look at the impact of pregabalin and gabapentin on functional assays of human μ receptor signaling. Gabapentin and pregabalin do not mediate their analgesic or toxic effects by a direct effect on the μ receptor. No allosteric modulation of μ receptor activity was identified in assays of GIRK, cAMP or ERK. High dose pregabalin caused increase in cAMP production, and the mechanism behind this warrants further investigation.

### Funding

Preeti Manandhar is supported by an International Research Excellence Scholarship from Macquarie University. The funders had no role in study design, data collection and analysis, decision to publish, or preparation of the manuscript.

### Grant Disclosures

The following grant information was disclosed by the authors:
Macquarie University.

### Competing Interests

Mark Connor is an Academic Editor for PeerJ. The other authors declare that they have no competing interests.

### Author Contributions

- Preeti Manandhar conceived and designed the experiments, performed the experiments, analyzed the data, prepared figures and/or tables, authored or reviewed drafts of the paper, and approved the final draft.
- Bridin Patricia Murnion conceived and designed the experiments, performed the experiments, analyzed the data, prepared figures and/or tables, authored or reviewed drafts of the paper, and approved the final draft.
- Natasha L. Grimsey conceived and designed the experiments, authored or reviewed drafts of the paper, and approved the final draft.
- Mark Connor conceived and designed the experiments, authored or reviewed drafts of the paper, and approved the final draft.
- Marina Santiago conceived and designed the experiments, analyzed the data, authored or reviewed drafts of the paper, and approved the final draft.

### Data Availability

Data used to plot CRC and dot plots using GraphPad prism are available in the Supplemental Files.

### Supplemental Information

Supplemental information for this article can be found online at http://dx.doi.org/10.7717/peerj.11175#supplemental-information.

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
