# Peer review of "Do gabapentin or pregabalin directly modulate the µ receptor?"

_PeerJ, doi:10.7717/peerj.11175_

## Round 0.1 · original submission · Major Revisions

Reviewer 2 has raised a very critical point about the mechanism of MOR desensitization that needs to be addressed before acceptance of this manuscript. The gabapentinoids can modulate MOR desensitization on prolonged agonist activation. As referenced by the authors there is a previous publication by another research group that suggests that pregabalin reverses tolerance to the respiratory depressant effect of morphine. Receptor desensitization may be a component of tolerance. Unfortunately the information provided in the present manuscript is not sufficient to support the claim that gabapentinoids do not modify MOR desensitization.

A number of kinases (e.g. GRKs/arrestins, PKC isoforms, ERKs and others as described in Williams et al 2012 Pharmacol Rev . 2013 Jan 15;65(1):223-54) have been implicated in MOR desensitization in different cell types and under different experimental conditions. The authors provide no information on the mechanism(s) underlying the desensitization they observe in AtT20 cells under their experimental conditions. Whilst they do show that the desensitization they observe is not altered by gabapentinoids they have not excluded the possibility that the mechanism responsible in their experiments is not the one that gabapentinoids interact with in vivo. For example, if the desensitization they observe is largely GRK/arrestin dependent then if the gabapentinoids inhibit PKC activity then no effect would have been observed (and vice versa). Thus, while the observation made by the authors is entirely valid, by not defining the mechanism(s) involved in the desensitization occurring under their experimental conditions it is difficult to interpret the observation.

Reviewer 1 ·

Basic reporting

Overall I think this is an important study despite the largely negative data presented as it in a solid way show that pregabalin and gabapentin does not have direct activation/modulation on the µ receptor. This is important knowledge given the increasing co-misuse of the drugs with opiates. The data and discussion is largely very well performed and described but I do have some suggestions on how to improve the study below.

Line 86-89: the reference for alternative drug targets is quite old (2006) and somewhat outdated as it has e.g. been shown that gabapentin does not act on GABA-B receptors. As the least, these two GABA-B publications (PMID: 11747901 & PMID: 12021399) should be added or better yet a never reference with updated information should be used instead.

Line 119: There are a number of different HEK293 cell lines available. Please specify which one you used.

Line 119-127: Please provide more details on how the stably transfected cell lines were generated. Which vectors/plasmids were used, which resistance gene did they contain, were the cell lines polyclones or single cell expansions and how was the stable expression confirmed?

E.g. line 196. In general, it would be more correct to term the graphs concentration-response curves as dose refer to a quantity of compound and not a concentration as depicted on the graphs.

Line 203-204: In most cases more than two conditions are compared in the statistical tests where it would then be more appropriate to use ANOVA test.

Line 221-231: To show specificity of the cell line and assay used please make a control experiment of morphine in non-transfected or mock-transfected HEK-293 and AtT20 cells.

Line 245: The abbreviation CRC is not explained. I assume it refers to concentration-response curve which would be fine. Please use the same nomenclature for these graphs througout the manuscript as previously noted for line 196.

Line 258-262: It is somewhat difficult to interpret Figure 4d as the increase in inverse BRET ratio is not put in perspective with the other graphs shown in the Figure 4a-c which have been normalized. I suggest to add the response of 10 µM morphine to Figure 4d which would make it possible to see of the increase in inverse BRET ratio caused by 100 µM pregabalin is a small or big effect compored to the morphine effects.

Line 267-268. The Figure 4e and 4f have been switched in the text compared to the figure.

Line 270-278: It would be appropriate to also test gabapentin and pregabalin alone (in absence of morphine) as has been done it the two other assays. This would also support the claim made in line 285.

Figure 5: It looks weird that -6.5 and -5.5 morphine was not tested as one would usually include more - not less - data points on the dynamic part of a CRC.

Line 317-329: I think that the finding of pregabalin potentiating forskoline (Figure 4f) is an interesting observation. But as mentioned in line 326-329 it is likely not mediated by the µ receptor or even Gs and the mechanism remains unknown. As the system used to test this effect is rather artificial compared to human therapeutic use I think line 317-329 is too speculative and I would rather note the observation and call for more studies to delineate the mechanism (as done in the conclusion) and determine if it has any human translation.

Otherwise, I think the discussion is sound but could elaborate on potential indirect mechanisms of pregabalin and gabapentin not explored in the present study such as changing cell surface µ receptor expression, synergistic action of pregabalin/gabapentin activity on voltage-gated ion-channels and downstream µ receptor signaling or activity of pregabalin/gabapentin on some of the heterodimeric receptors which the µ receptor has been reported to form with e.g. Kappa and delta receptors and recently also GPR139.

Experimental design

See above.

Validity of the findings

See above.

Additional comments

See above.

Reviewer 2 ·

Basic reporting

The topic is an important one. This manuscript by investigates potential interactions between the gabapentinoids (gabapentin and pregabalin) with the mu-opioid receptor (MOR). This is an important area of investigation as it has clinical as well as drug overdose death implications. Gabapentinoids have become drugs of abuse and while they are not normally lethal in overdose by themselves they have become associated with heroin overdose deaths. The manuscript is well written and the citing of the relevant scientific literature is appropriate.

Experimental design

In this manuscript the authors report that the gabapentinoids in low and high doses do not themselves activate MOR nor do they potentiate the signalling events induced by morphine i.e. they do not act as positive allosteric modulators. These experiments have been carefully performed using the human MOR expressed in two cell line, HEK293 and AtT20 cells, the later being pituitary cells with some neuronal phenotype. Both cell types have been extensively used in previous studies of MOR function by the authors and other researchers. The assays used are well established for studies of MOR function

Validity of the findings

The authors also investigate whether the gabapentinoids can modulate MOR desensitization on prolonged agonist activation. As referenced by the authors there is a previous publication by another research group which suggests that pregabalin reverses tolerance to the respiratory depressant effect of morphine. Receptor desensitization may be a component of tolerance. Unfortunately the information provided in the present manuscript is not sufficient to support the claim that gabapentinoids do not modify MOR desensitization.
A number of kinases (e.g. GRKs/arrestins, PKC isoforms, ERKs and others as described in Williams et al 2012 Pharmacol Rev . 2013 Jan 15;65(1):223-54) have been implicated in MOR desensitization in different cell types and under different experimental conditions. The authors provide no information on the mechanism(s) underlying the desensitization they observe in AtT20 cells under their experimental conditions. Whilst they do show that the desensitization they observe is not altered by gabapentinoids they have not excluded the possibility that the mechanism responsible in their experiments is not the one that gabapentinoids interact with in vivo. For example, if the desensitization they observe is largely GRK/arrestin dependent then if the gabapentinoids inhibit PKC activity then no effect would have been observed (and vice versa). Thus, while the observation made by the authors is entirely valid, by not defining the mechanism(s) involved in the desensitization occurring under their experimental conditions it is difficult to interpret the observation.

Additional comments

No comment

---

## Round 0.2 · accepted · Accept

Dear Dr. Santiago,

Thank you for your submission to PeerJ.
I am writing to inform you that your manuscript - Do gabapentin or pregabalin directly modulate the µ receptor? - has been Accepted for publication. Congratulations!

Congratulations again, and thank you for your submission.

With kind regards,
Durga Tripathi
Academic Editor, PeerJ

Reviewer 1 ·

Basic reporting

no comment

Experimental design

no comment

Validity of the findings

no comment

Additional comments

The authors have addressed my comments in my review of the first submission in an adequate way and I thus have no further comments.

Reviewer 2 ·

Basic reporting

Fine

Experimental design

Fine

Validity of the findings

fine

Additional comments

The revisions made bt the authors are adequate to meet my original reservations.